# Leveraging Entanglement Entropy for Deep Understanding of Attention Matrix in Text Matching

## Abstract

The formal understanding of deep learning has made great progress based on quantum many-body physics. For example, the entanglement entropy in quantum many-body systems can interpret the inductive bias of neural network and then guide the design of network structure and parameters for certain tasks. However, there are two unsolved problems in the current study of entanglement entropy, which limits its application potential. First, the theoretical analysis of entanglement entropy was only investigated in the representation of a single object (e.g., an image or a sentence), but has not been well studied in the matching of two objects (e.g., question-answering pairs). Second, the entanglement entropy can not be qualitatively calculated since the exponentially increasing dimension of the matching matrix. In this paper, we are trying to address these two problem by investigating the fundamental connections between the entanglement entropy and the attention matrix. We prove that by a mapping (via the trace operator) on the high-dimensional matching matrix, a low-dimensional attention matrix can be derived. Based on such a attention matrix, we can provide a feasible solution to the entanglement entropy that describes the correlation between the two objects in matching tasks. Inspired by the theoretical property of the entanglement entropy, we can design the network architecture adaptively in a typical text matching task, i.e., question-answering (QA) task. Our adaptive strategy sets a new state-of-the-art performance in TREC-QA task, with 2.9% absolute improvement over a recent model.

## 1 Introduction

Fundamental connections between neural network and quantum mechanics have been built (Carleo & Troyer, 2017; Levine et al., 2018; Cai & Liu, 2018). Neural networks are adopted to solve the quantum many-body problem (Carleo & Troyer, 2017; Cai & Liu, 2018), while quantum many-body function has been involved to explain the expressive ability of the neural networks (Levine et al., 2018). The formal understanding of deep neural network has made great progress based on quantum many-body physics. Levine et al. (2019) summarize the theoretical concepts and properties of quantum entanglement in different neural network architectures. Specifically, for recurrent networks, the Start-End separation rank as a method of measuring quantum entanglement, in order to quantitatively describe the depth effectiveness for modeling long-term memory capacity (Levine et al., 2017; 2019). For deep convolutional network, Levine et al. (2017; 2019), shows that the entanglement entropy can help us understand the inductive bias of the neural network. Such understanding also provide a guidance of how to control the inductive bias of the designed network, via the network parameters (e.g., channel numbers).

However, there are still two limitations in the current study (Levine et al., 2017; 2019; Zhang et al., 2018b). First, the quantum entanglement measure (i.e., the entanglement entropy) is considered as the indicator of the network's expressive ability, but such an indicator only reflects the intricate correlation structures of each single input object (e.g., an image or a text). In other words, previous theoretical analyses are conducted only for the *representation* of a single object. However, for the

*matching* problem, the theoretical analysis should be extended, in order to model the correlation distributions between two objects (e.g., question-answering pairs).

On the other hand, the quantitative calculation of the entanglement entropy is also a problem. In theory, the entanglement entropy has a relation with the network parameters reflecting the inductive bias of the network. However, in practice, the entanglement entropy remains infeasible to calculate. This is due to the fact that the tensor product occurs in the quantum many-body function for representing the image and text (Levine et al., 2017; Zhang et al., 2018b), and the dimension of the matching matrix representing the quantum entanglement between two systems will increase exponentially. This leads to the difficulty in solving the entangled entropy by the singular value decomposition (SVD) of such a high-dimensional matching matrix.

In this paper, we are trying to address these two problem by investigating the fundamental connections between the entanglement entropy and the attention matrix. Under certain conditions and mappings, we demonstrate that a low-dimensional attention matrix can encode certain information (e.g., the trace of the block matrix) of the high-dimensional matching matrix. Then based on the low-dimensional attention matrix, we can quantitatively calculate entanglement entropy. Specifically, we can represent the high-dimensional matching matrix by block matrices, and then compute the trace for each block. We will show that the resulting matrix is equivalent to the attention matrix based on the cosine distance calculation. By the SVD decomposition of this matrix, the singular values are obtained to solve the entanglement entropy quantitatively.

In our work, inspired by (Levine et al., 2018), we can adaptively design convolutional network based on the calculable entanglement entropy. Specifically, for the more complex the inputs (i.e., the long-range correlations), more kernels should be assigned in the relatively-deeper layers, while more kernels should be assigned in the relatively-shallower layers for short-range correlations. Intuitively, in text matching task like question answering system, the short-range correlations can refer to some simple question-answering pairs with many common words between question and answer sentences, which can be matched locally by some overlapping features (e.g., the statistics of a single word or nearby word combinations like N-gram). While long-range correlations refer to the question-answering pairs with less common words which their effective matching may need higher-level semantic information extracted from a global context. This strategy significantly improves the final performance on two typical text matching tasks called TREC-QA and YAHOO-QA. In particular,our model sets a new state-of-the-art performance in TREC-QA, with 2.9% absolute improvements.

## 2 BACKGROUND AND BASIC NOTATIONS

Since our work is mainly for the text matching task of a sentence pair, we briefly introduce a recent Quantum Many-body Wave Function inspired Language Modeling ( QMWF-LM) (Zhang et al., 2018b). In QMWF-LM, a state can be represented by a Dirac notation $|w\rangle$, which can be considered as a column vector $\boldsymbol{w}$. For better understanding, in our paper, we will use a vector $\boldsymbol{w}$ to represent a Dirac vector $|w\rangle$. A word can be described by a state vector as follows.

$$\boldsymbol{w} = \alpha_1 \boldsymbol{e}_1 + \ldots + \alpha_m \boldsymbol{e}_m \tag{1}$$

where $\{\boldsymbol{e}_1, \ldots, \boldsymbol{e}_m\}$ is a set of base vectors. Each $\alpha_i (i \in 1, \ldots, m)$ is a probability amplitude and $\sum_i^m \alpha_i^2 = 1$. In practice, $\boldsymbol{w}$ can be the distributive representation or the one-hot representation for the word $w$. A sentence can be represented by tensor product $\otimes$ [1] among words in a sentence. For the representation of a sentence $S$ with $n$ words, it can be written as follows,

$$\boldsymbol{s} = \boldsymbol{w}_1 \otimes \ldots \otimes \boldsymbol{w}_n \tag{2}$$

where $\{\boldsymbol{w}_1 \ldots \boldsymbol{w}_n\}$ are the word which is defined in E.q. 1.

In order to establish the correlation between two sentences ($V_Q$ with a number (saying $a$) of words in the question and $V_A$ with $b$ words in the answer), we can have the following equation. The state of the two sentences system can be defined as:

$$\boldsymbol{\psi} = \sum_{i=1}^{m^a} \sum_{j=1}^{m^b} \boldsymbol{T}_{ij} \boldsymbol{\phi}_i^Q \otimes \boldsymbol{\phi}_j^A \tag{3}$$

---

[1]e.g., $\boldsymbol{e}_1 = (1,0)^T, \boldsymbol{e}_2 = (0,1)^T, \boldsymbol{e}_1 \otimes \boldsymbol{e}_2 = (0,1,0,0)^T$

where the state vector $\psi$ is a composite system between two sentences (i.e., $V_Q$ and $V_A$), $\phi_i^Q$ is a high-dimensional ($m^a$, $m$ is the dimension of word vector) base vector for the sentence $V_Q$, $\phi_i^Q$ also is a high-dimensional ($m^b$) base vector for the sentence $V_A$. Especially, $T_{i,j}$ is the element for the high-dimensional ($m^a$ by $m^b$) matching matrix $T$.

## 3 A FEASIBLE METHOD FOR QUANTIFYING ENTANGLEMENT ENTROPY

In this section, firstly, we use entanglement entropy to measure the quantum entanglement describing the correlation between two subsystems (e.g.,a sentence pair). Secondly, we prove the equivalence between the quantum entanglement and the attention matrix under certain conditions. Finally, based on the attention matrix, the entanglement entropy is quantitatively calculated, which facilitates us to design the neural network architecture through the overall inductive bias.

### 3.1 ENTANGLEMENT ENTROPY IN TEXT MATCHING

Entanglement entropy is used to represent correlations between two subsystems in deep neural network (Levine et al., 2018; 2019). Specifically, it requires a matrix to model all the quantum entanglement of the two subsystems, and then the matrix is decomposed to obtain the singular values, which often correspond to the important information hidden in the matrix, and the importance is positively correlated with the singular value size. Finally, based on the singular values, the entanglement entropy representing the degree of entanglement between the two subsystems can be obtained. In the matching task, we use quantum many-body wave functions (Zhang et al., 2018b) as the basic language representation to get the entanglement entropy between sentence pairs.

In matching tasks (e.g., Q&A task), a question-answering pair can be considered as two subsystems, $V_Q = \{w_1^Q, \ldots, w_a^Q\}$, $V_A = \{w_1^A, \ldots, w_b^A\}$. $V_Q \in \mathcal{H}^Q$ with dimension $m^a$ and $V_A \in \mathcal{H}^A$ with dimension $m^b$, respectively. $m$ is the dimension of word vector. The composite system $\psi_S$ of $V_Q$ and $V_A$, which can be written as (Levine et al., 2018; Cohen & Shashua, 2016):

$$\psi_S = \sum_{i=1}^{m^a} \sum_{j=1}^{m^b} T_{i,j} \phi_i^Q \otimes \phi_j^A \tag{4}$$

where $\phi_i^Q (i \in \mathcal{H}^Q)$ and $\phi_j^A (j \in \mathcal{H}^A)$ are basis vectors of $V_Q$ and $V_A$, respectively. The dimensions of basis vectors about $V_Q$ and $V_A$ are $m^a$ and $m^b$, respectively. All the entries $T_{i,j}$ shows the correlation between $\phi_i$ and $\phi_j$. When $i = 1$, $j$ is taken from 1 to $m^b$, $\phi_1^Q$ represents the basis vector of $i = 1$ in the tensor space $\mathcal{H}^Q$ (each base vector represents a certain combined semantics of all words), $\{\phi_j^Q\}_{j=1}^{m^a}$ represents all base vectors in the tensor space $\mathcal{H}^Q$, $T_{1,j}$ represents the corresponding probability amplitude of the base vector $\phi_1^Q$ of the $V_Q$ subsystem combined with the base vector $\phi_j^A$ of the $V_A$ subsystem. The matching matrix $T$ can obtain all probability amplitude distributions of $i$ from 1 to $m^a$ in parallel.

The matching matrix $T$ contains all the probability amplitude distributions between the two subsystem basis vectors. Singular value decomposition (SVD) (Schollwoeck & White, 2013) on the matching matrix $T$ can be written:

$$T = \sum_{i=1}^{r} \lambda_i \varphi_{Q_i} \cdot \varphi_{A_i}^T \tag{5}$$

where $\varphi_{Q_i}$ and $\varphi_{A_j}$ are $r$ vectors in new bases for $\mathcal{H}^Q$ and $\mathcal{H}^A$, respectively. $\lambda_1 \geq \ldots \geq \lambda_r > 0$ are the singular values, which often correspond to the important information hidden in the matrix, and the importance is positively correlated with the singular value size, the size of the singular value represents the weight of the current basis vector of the two subsystems. $\lambda_i$ is also understood as probability amplitude, $\sum_i \lambda_i^2 = 1$. $\lambda_i$ represents the weight of principal semantic bases between $V_Q$ and $V_A$, and reflects the weight of effective semantics to some extent (since it is larger than zero). Entanglement entropy (Yo, 2015) is computed as follows on the matching matrix $T$,

$$S = -\sum_{i=1}^{r} |\lambda_i|^2 ln |\lambda_i|^2 \tag{6}$$

where $S \in (0, ln\,(r))$, $S_{\max} = ln\,(r)$, $r$ is the number of non-zero singular values, and the number of $r$ is called Schmidt number (Shi et al., 2006; Levine et al., 2018). The entanglement entropy describes the correlation between two systems, and the Schmidt number indicates the upper bound of the network expression ability and can assert the role of channel numbers of each layer in the overall inductive bias (Levine et al., 2018). Due to this dimensional catastrophe [2], it is infeasible to calculate entanglement entropy and Schmidt number. This leads us to practically apply inductive bias is limited in the experiment. However, in matching tasks, an attention matrix often is used to describe correlation between two systems. Based on the above ideas, we propose a claim to calculate the entanglement entropy.

**Claim 1.** *Assume the compound relation of word vectors in $V_Q$ (or $V_A$) are considered in the case of $1$-order, and it just is considered for the compound relation between each word vector in $V_Q$ and each word vector in $V_A$. An attention matrix can be received by computing the trace of block matrices of the matching matrix $\boldsymbol{T}$.*

*Proof.* Assume the compound relation of word vectors in $V_Q$ (or $V_A$) are not consider (e.g., 1-order), the dimensions of $\mathcal{H}^Q$ and $\mathcal{H}^A$ in E.q. 4 are $a \times m$ ($am$) and $b \times m$ ($bm$), respectively. E.q. 4 can be rewritten as follow:

$$\boldsymbol{\psi}_S = \sum_{i=1}^{am} \sum_{j=1}^{bm} \boldsymbol{T}_{i,j} \boldsymbol{\phi}_i^Q \otimes \boldsymbol{\phi}_j^A \tag{7}$$

where $\boldsymbol{\phi}_i^Q$ is the base vector from a group of complete base vector space $\boldsymbol{\phi}_1^Q, \ldots, \boldsymbol{\phi}_{am}^Q$, and $\boldsymbol{\phi}_j^A$ is the base vector from a group of complete base vector space $\boldsymbol{\phi}_1^A, \ldots, \boldsymbol{\phi}_{bm}^A$. The $\boldsymbol{T}$ is a matching matrix of $am \times bm$. Divide the matching matrix $\boldsymbol{T}$ into $a \times b$ block matrices $\{\boldsymbol{P}_{ij} \in \mathbb{R}^{m \times m}\}(i \in [a]^3; j \in [b])$, which can be written as $\boldsymbol{P}_{ij} = w_i^Q \otimes w_j^A$. Each element $\boldsymbol{E}_{ij}$ of the attention matrix $\boldsymbol{E}$ is equal to the inner product of two word vectors which are from the sets $V_Q$ and $V_A$, which is $\boldsymbol{E}_{ij} = \boldsymbol{w}_i^Q \cdot \boldsymbol{w}_j^{A^T} (i \in [a]; j \in [b])$. There is a mathematical connection between $\boldsymbol{P}_{ij}$ and $\boldsymbol{E}_{ij}$, which can be written as:

$$\boldsymbol{E}_{ij} = trace(\boldsymbol{P}_{ij}) \tag{8}$$

The relation between the matching matrix $\boldsymbol{T}$ and the attention matrix $\boldsymbol{E}$ has been shown as the Fig.1.

$\square$

Figure 1: The relation between the attention matrix $\boldsymbol{E}$ and the matching matrix $\boldsymbol{T}$: $P_{ij}$ is the block matrix from the matching matrix $\boldsymbol{T}$, $\boldsymbol{E}_{ij}$ is equal to the trace of matrix $\boldsymbol{P}_{ij}$. $\boldsymbol{T}_{k,l}^{i,j}$ ($k, l \in [m], i \in [a], j \in [b]$) is the entry of block matrix $\boldsymbol{P}_{ij}$, the trace represent the summation of diagonal elements of matrix $\boldsymbol{P}_{ij}$.

---

[2]Dimensional catastrophe is mainly reflected in two aspects. First, in Eq.4, the matrix $\boldsymbol{T}(\boldsymbol{T} \in \mathbb{R}^{m^a \times m^b})$ has dimensional catastrophes. Second, even if the SVD is decomposed on the matrix $\boldsymbol{T}$, its number of singular values still has dimensional catastrophe (i.e., number of singular values $\leq min(m^a, m^b)$).

[3][a] represents a set $\{1, 2, 3, \ldots, a\}$

Thus, we use the attention matrix $\boldsymbol{E}$ instead of the matching matrix $\boldsymbol{T}$ to quantify the entanglement entropy. In our **Conjecture 1**, we provide a generalized proof of the mathematical relationship between the attention matrix and the entanglement properties. In the experiments, we use the cosine distance about two vectors to replace the inner product actually. In the next subsection, we will describe our Algorithm.

### 3.2 NETWORK DESIGN BASED ON ENTANGLEMENT ENTROPY

Recall that the sentence pair contained query sentence and answer sentence can be represented through two sets, which are $V_Q$ and $V_A$, respectively. where $V_Q = \{\boldsymbol{w}_1^Q, \ldots, \boldsymbol{w}_a^Q\}$, $V_A = \{\boldsymbol{w}_1^A, \ldots, \boldsymbol{w}_b^A\}$, and $a + b = n$.

Based on the **Claim 1**, the attention matrix between of $V_Q$ and $V_A$ can be computed by using *cosine* similarity in experiment,

$$\boldsymbol{E}_{ij} = \frac{\boldsymbol{w}_i^Q \cdot \boldsymbol{w}_j^A}{\|\boldsymbol{w}_i^Q\| \times \|\boldsymbol{w}_j^A\|} \tag{9}$$

the *attention matrix* $\boldsymbol{E} \in \mathbb{R}^{a \times b}$ can be decomposed by SVD.

After that, the entanglement entropy can is calculated through E.q., 6 on attention matrix $\boldsymbol{E}$. while the upper bound of entanglement entropy $S_{max}$ is obtained based on the number of singular values. The normalized value of the entanglement entropy is used in practice, it can be written as follows.

$$D = (S_{max} - S)/S_{max}. \tag{10}$$

We calculate all data sample differences $D$, and then find the median $D_m$ of the difference $D$. The samples with $D$ greater than $D_m$ are defined as the group with the short-range correlations, while the samples less than $D_m$ are defined as the group with long-range correlations. Based on the connection between deep learning and quantum entanglement (Levine et al., 2018), i.e., deep learning describes the correlation required for deep learning tasks, we can adaptively design convolutional network by the correlation (e.g., long/short-range correlation). For the more complex the input (i.e., the long-range correlation), more kernels should be assigned in the relatively-deeper layers, while more kernels should be assigned in the relatively-shallower layers for short-range correlations. In our opinion, quantitatively calculating entanglement entropy by attention matrix can not only help us adaptively design the network architecture, but also provides a new perspective for us to understand the attention matrix in matching tasks.

## 4 EXPERIMENTS

Question Answering aims to rank the answers from a candidate answer pool for a given question. The ranking is usually based on two parts, namely, **representation part** for a single sentence and **matching part** between these two representations.

*Datasets* We validated our model through the TREC-QA (Wang et al., 2007) and YAHOO-QA (Tay et al., 2017) datasets in the matching task. TREC-QA dataset contains two training sets, namely TRAIN and TRAIN-ALL. We use TRAIN-ALL, which is larger and contains noisier question-answering pairs, in order to verify the robustness of the proposed model. YAHOO-QA dataset is collected from yahoo answers for community-based question answering. The answers are generally longer than those in TREC-QA. As introduced in this paper (Tay et al., 2017), we select the QA pairs containing questions in which the token number of its corresponding answers is more than 5 and less than 59 after removing non-alphanumeric characters. For each question, we construct negative samples by ranking the top 4 answers from the whole answer sentences according to BM25.

*Parameters Setting* We utilize the Adam (Bengio & LeCun, 2015) optimizer with learning rate 0.001 and use the best model obtained in the dev dataset for evaluation in the test set. The batch size and $L2$ regularization are set to $64$ and $0.00001$, respectively. The 50-dimension word embeddings are trained by word2vec (Burges et al., 2013) on English Wikimedia dump, in which the Out-of-Vocabulary words are randomly initialized by a uniform distribution in the range of $(-0.25, 0.25)$.

Table 1: Results on the statistics of dataset. $\alpha$, $\beta$ and $\varepsilon$, $\delta$ denote significant improvement (with $p < 0.05$) over QLM (Sordoni et al., 2013) and NNQLM-II (Zhang et al., 2018a), QMWF-LM (Zhang et al., 2018b), CNM (Li et al., 2019) respectively, according to Wilcoxon signed-rank test.

| | TREC-QA | | YAHOO-QA | |
|---|---|---|---|---|
| | MAP | MRR | p@1 | MAP |
| QLM (Sordoni et al., 2013) | 0.678 | 0.726 | 0.395 | 0.604 |
| NNQLM-II (Zhang et al., 2018a) | $0.759^{\alpha}$ | $0.825^{\alpha}$ | $0.466^{\alpha}$ | $0.673^{\alpha}$ |
| QMWF-LM (Zhang et al., 2018b) | $0.752^{\alpha}$ | $0.814^{\alpha}$ | $0.575^{\alpha,\beta}$ | $0.745^{\alpha,\beta}$ |
| CNM (Li et al., 2019) | $0.770^{\alpha,\beta,\varepsilon}$ | $0.859^{\alpha,\beta,\varepsilon}$ | — | — |
| **Our-Model** (adaptive setting for kernels) | $\mathbf{0.881}^{\,\alpha,\beta,\varepsilon,\delta}$ | $\mathbf{0.924}^{\alpha,\beta,\varepsilon,\delta}$ | $\mathbf{0.630}^{\alpha,\beta,\varepsilon}$ | $\mathbf{0.789}^{\alpha,\beta,\varepsilon}$ |
| deep-more-kernels-fixed for whole dataset | 0.668 | 0.762 | 0.342 | 0.574 |
| shallow-more-kernels-fixed for whole dataset | 0.665 | 0.759 | 0.346 | 0.557 |
| Multi-Perspective CNN (Rao et al., 2016) | 0.780 | 0.834 | — | — |
| Attention Pooling Networks (dos Santos et al., 2016) | — | — | 0.560 | 0.726 |
| Holographic Dual LSTM (Tay et al., 2017) | 0.752 | 0.815 | 0.557 | 0.735 |
| Cross Temporal Recurrent Networks (Tay et al., 2018) | 0.771 | 0.838 | 0.601 | 0.755 |
| Multi-Perspective CNN + QC (Bender et al., 2018) | 0.836 | 0.863 | — | — |
| QC + RNN + Pre-Attention (Kamath et al., 2019) | 0.852 | 0.891 | — | — |

## 4.1 EVALUATION METRICS

TREC-QA is evaluated by MAP(mean average precision), MRR(mean reciprocal rank), while YAHOO-QA is evaluated in term of P1 (precision at one) and MRR since it targets more on the first ranking item.

## 4.2 CONFIGURATIONS FOR OUR MODEL

In this paper, we adopt the quantum many-body wave functions (Zhang et al., 2018b) as the basic representation components, to represent both question and answers sentence as two subsystems i.e. $Q$ and $A$ respectively. Since the whole system with question-answering pairs is composed by these two subsystems, the entanglement entropy is adopted to quantify correlations between subsystems.

Generally speaking, there are typically two kinds of correlations in question answer task, namely long-range correlation and short-range correlation. The former refers to more complex matching with consideration of the global sentence. The latter refers to the cases that there are already overlapping words or phrases between the question sentence and the answer sentence, which can be matched locally at the lexical and syntactic level. As discussed in our algorithm, the entanglement entropy is a quantitative metric to distinguish how much semantically entangled the question and answer are. Thus we could divide question-answering pairs into two sub-datasets, i.e. long-range sub-dataset and short-range sub-dataset. Then, we can design an *adaptive setting for kernels* for different sub-datasets. For long-range sub-dataset, we should choose the more kernels in deep layers (namely deep-more-kernels mode). while for short-range sub-dataset, we should choose the more kernels in the shallow layers (namely shallow-more-kernels mode). At last, we combine the evaluation scores of the long/short range sub-datasets as the data for the whole dataset to be evaluated.

We set a parameter $r$ from following modes for an input layer and three-layer CNN network:

(1) shallow-more-kernels: [m,3r,2r,m]
(2) deep-more-kernels:[m,2r,3r,m]

$m$ is a hyper-parameter for the kernel count of the last CNN layer, which is set as dimension of word embedding for guaranteeing the same parameter scale between these two modes. The parameter count for both configurations is identical: $m \cdot 3r + 3r \cdot 2r + 2r \cdot m = 6r^2 + 5mr$. Shallow-more-kernels mode has more kernels in the lower layers while the deep-more-kernels mode has more kernels in higher layers. The code will be open-sourced in [4].

**Comparative Setup** To validate the above adaptive setting for parameters, we design two comparative fixed settings without dividing the dataset. Fixed setting 1: the deep-more-kernels parameter mode is used for the whole dataset. Fixed setting 2: the shallow-more-kernels mode is applied to the

---

[4]https://github.com/anonymous/anonymous.git

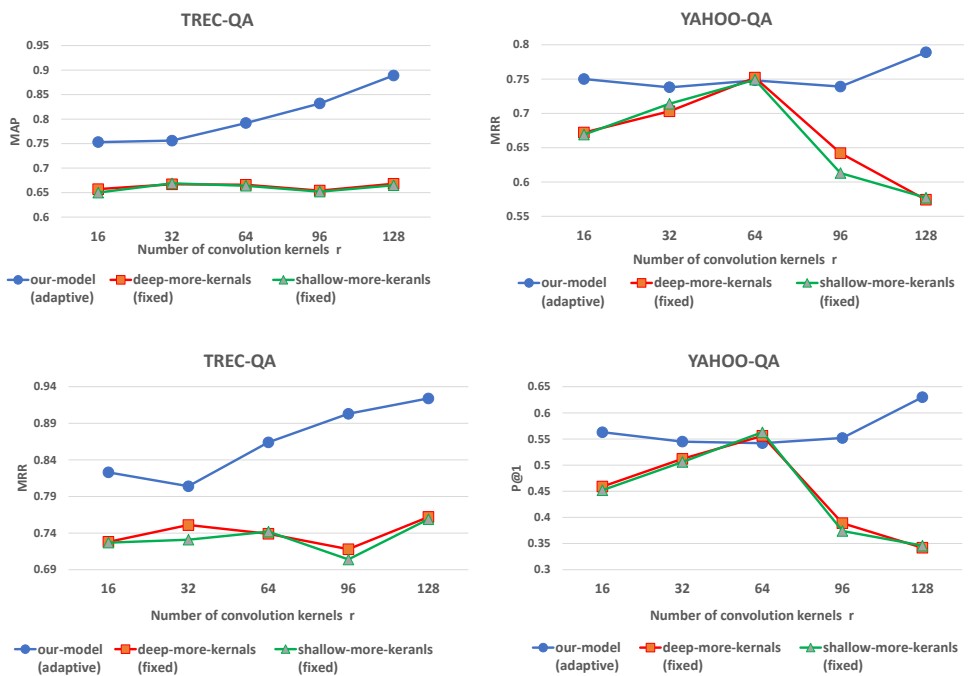

Figure 2: We compare our model (adaptive settings for kernels) to two fixed modes, deep-more-kernels for the whole dataset and shallow-more-kernels for the whole dataset. For TREC-QA, our model with an adaptive setting for kernels achieved significant improvements on MAP and MRR (with $p < 0.01$) over the fixed settings. For YAHOO-QA, we can get similar results. Both the statistic tests are based on Wilcoxon signed-rank test.

whole dataset. Except for the setting for the kernel numbers and dataset paritition, Other parameters in both adaptive and fixed settings are the same, for a fair comparison.

### 4.3 RELEVANT METHODS FOR COMPARISON

**QLM** (Sordoni et al., 2013) adopts the density matrix of quantum mechanics to model the dependencies between words such as query words and documents.

**NNQLM-II** (Zhang et al., 2018a) is a end-to-end quantum-like language model for question answering. It adopts a convolution neural network over the multiplication between two density matrices.

**QMWF-LM** (Zhang et al., 2018b) is based on the quantum many-body wave function representation and a further convolution neural network for an approximate tensor decomposition.

**CNM** (Li et al., 2019) utilizes the probability properties of the density matrix, indirectly measures the distance between two density matrices through a trainable measurement operation, and uses training data to provide flexible matching scores.

Our model is also compared to a series of CNN and RNN models (Rao et al., 2016; Tayyar Madabushi et al., 2018; Tay et al., 2017; 2018; Kamath et al., 2019). To the best of our knowledge, Question Classification + RNN + Pre-Attention (Kamath et al., 2019) is the state of art in the raw version of TREC-QA. [5].

### 4.4 EXPERIMENTAL RESULTS

Tab. 1 reports the results on TREC-QA dataset and YAHOO-QA dataset. The first group shows a comparison of between our model with four quantum-inspired language models. For TREC-QA, our model significantly outperforms QLM by 20.3% on MAP and 19.8% on MRR, respectively. Compared with NNQLM-II and QMWF-LM, CNM, it also reflects the superiority of our model.

---

[5]https://aclweb.org/aclwiki/Question_Answering_(State_of_the_art)

For YAHOO-QA, our model is significantly better than QLM 23.5% and 18.5% in $P@1$ and MRR, significantly better than NNQLM-II 16.4% and 11.6% in $P@1$ and MRR, and better than QMWF-LM 5.5% and 4.4% in $P@1$ and MRR. In summary, compared with three quantum-inspired language models, our model shows good experimental performance on both tasks.

In the second group, our model with adaptive setting for kernels is compared with other two fixed settings, i.e., deep-more-kernels fixed for the whole dataset and shallow-more-kernels for the whole dataset (see Sec. 4.2 for details). For TREC-QA, the results show that both MAP and MRR are at least 15% better than them. For YAHOO-QA, the results show that both $P@1$ and MRR are at least 20% better than the comparative setup. In summary, the results reflect the effectiveness of the adaptive settings of parameters for two divided data-subsets.

In the third group, we compare our model with a number of CNN-based models and RNN-based models (Rao et al., 2016; Tayyar Madabushi et al., 2018; Tay et al., 2017; 2018). Multi-Perspective CNN (Rao et al., 2016) and Multi-Perspective CNN +PairwiseRank + Question classification (Tayyar Madabushi et al., 2018) model the QA task as a ranking task and proposes a pairwise ranking method that can directly utilize the existing point-by-point neural network model as a basic component. HD-LSTM (Tay et al., 2017) and CTRN (Tay et al., 2018), Question Classification + RNN + Pre-Attention (Kamath et al., 2019) propose an RNN-based approach to match question-answering pairs. It should be emphasized that our model is better than Question Classification + RNN + Pre-Attention (Kamath et al., 2019), which is the state of art in the raw version of TREC-QA. For TREC-QA and YAHOO-QA, our models are better than the results (Rao et al., 2016; Tayyar Madabushi et al., 2018; Tay et al., 2017; 2018; Kamath et al., 2019) reported in the original paper.

## 4.5 DISCUSSION AND ANALYSIS

**Analysis of Adaptive Settings** In Fig. 2, we investigate the difference between the adaptive setting in our model and two fixed settings introduced. Recall that in our model, we first identify the correlation between sentence pairs based on the attention matrix and entanglement entropy. Then, according to the correlation, the question-answering pairs could be divided into two sub-dataset, i.e, long-range sub-dataset and short-range datasets. Therefore, we can *adaptive set the kernels for different sub-dataset*. On the other hand, the two fixed settings apply the same parameter setup (deep-more-kernels or shallow-more-kernels) for the whole dataset.

As we can see from Fig. 2 TREC-QA, in our model, with the increase of the parameter $r$, MAP, MRR and $P@1$ show an upward trend. For the fixed settings, i.e., deep-more-kernels for the whole dataset and shallow-more-kernels for the whole dataset, it does not show better performance as $r$ changes, and even show a downward trend. This suggests that our *adaptive kernels settings* can well capture the language matching characteristics of the question-answering pairs, and the alternative approach to the entangle entropy as the measurement for the language correlations, work well for dividing the dataset into two subsets, reflecting different ranges of language correlations.

**Influence of Channels in Convolution** In Fig. 2, as $r$ grows, the MRR and MAP of TREC-QA and YAHOO-QA increases. We select the optimal $r$ in a range $[16, 32, 64, 96, 128]$. For different datasets, we set a same number of channels to obtain the final performance. In our model, for the TREC-QA and YAHOO-QA, when the parameter $r$ is set to 128, the experimental results as shown in Tab. 1 can be obtained.

**Efficiency Analysis** we utilize adaptive settings of kernels for different sub-datasets. The efficiency relies on the deep convolution neural network. In our experiment, for TREC-QA, the training epoch is set to be 100, while for YAHOO-QA, after training 30 epochs, we will obtain the results.

## 5 CONCLUSIONS AND FUTURE WORK

In this paper, we aim to extend previous studies on the theoretical and practical use of the concept of quantum entanglement in neural network. We demonstrate the connection between the entanglement entropy and the attention matrix in text matching tasks. This allows us to quantitatively calculate the entanglement entropy, which helps us adaptively setting network structures and parameters, and achieve effective performance on two typical QA tasks. In future work, we will investigate the entanglement entropy under high-order conditions and in other networks, e.g., recurrent networks.

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

## A  ATTENTION MATRIX AND ENTANGLEMENT ENTROPY

Given a sentence pair with word embedding $\{\boldsymbol{w}_i \in \mathbb{R}^m\}(i \in [n])$, which can be represented using two sets, i.e., $V_Q = \{\boldsymbol{w}_1^Q, \ldots, \boldsymbol{w}_a^Q\}$, $V_A = \{\boldsymbol{w}_1^A, \ldots, \boldsymbol{w}_b^A\}$.

**Conjecture 1.** *Assume the compound relation of word vectors in $V_A$ are considered in the case of $n$-order, and it just is considered for the compound relation between each word vector in $V_Q$ and $b$ words vectors in $V_A$. The attention matrix can be received by computing the trace of block matrices of matrix $\boldsymbol{T}$.*

*Proof.* Firstly, the mathematical relationship between the attention matrix and the quantum entanglement matrix is proved under the 1-order condition in 3.1.

Assume the compound relation of word vectors in $V_A$ are considered in the 2-order, the dimension of $\mathcal{H}^Q$ is $a \times m$ ($am$) and $\mathcal{H}^A$ is $C_b^2 \times m^2$, i.e., $C_b^2 m^2$ ($C_b^2$ is a Combination number[6]). E.q. 4 can be rewritten as follow:

$$\boldsymbol{\psi}_S = \sum_{i=1}^{am} \sum_{j=1}^{C_b^2 m^2} \boldsymbol{T}_{i,j} \boldsymbol{\phi}_i^Q \otimes \boldsymbol{\phi}_j^A \tag{11}$$

where $\{\boldsymbol{\phi}_i^Q\}_{i=1}^{am}$ and $\{\boldsymbol{\phi}_j^A\}_{j=1}^{C_b^2 m^2}$ are base vectors for $\mathcal{H}^A$ and $\mathcal{H}^Q$. The $\boldsymbol{T}$ is a martix of $am \times C_b^2 m^2$. Divide the $\boldsymbol{T}$ into $a \times C_b^2$ block matrices $\{\boldsymbol{P}_{ij} \in \mathbb{R}^{m \times m^2}\}(i \in [a]; j \in [C_b^2])$, which can be written as $\boldsymbol{P}_{ij} = \boldsymbol{w}_i^Q \otimes \boldsymbol{w}_j^A$, $\boldsymbol{w}_j^A$ ($\boldsymbol{w}_j^A \in \mathbb{R}^{m^2}$) represents the compound semantics of any two-word vectors in $V_A$. Matrix $\boldsymbol{P}_{ij}$ can be regarded as a 3-order tensor $\boldsymbol{P}_{t_1 t_2 t_3}$ ($\boldsymbol{P}_{t_1 t_2 t_3} \in \mathbb{R}^{m \times m \times m}$), each element $\boldsymbol{E}_{ij}$ ($i \in [a]; j \in [C_b^2]$) of the attention matrix $\boldsymbol{E}$ is equal to the sum of the 3-order tensor $\boldsymbol{P}_{t_1 t_2 t_3}$ diagonal elements, i.e., $\boldsymbol{E}_{ij} = <\boldsymbol{w}_{t_1}^Q, \boldsymbol{w}_{t_2}^A \cdot \boldsymbol{w}_{t_3}^A> (t_1 \in [a], t_2 \neq t_3$ and $t_2, t_3 \in [b])$. there is a mathematical connection between $\boldsymbol{P}_{t_1 t_2 t_3}$ and $\boldsymbol{E}_{ij}$, which can be written as:

$$\boldsymbol{E}_{ij} = trace(\boldsymbol{P}_{ij}) \tag{12}$$

Thus, we use the attention matrix $\boldsymbol{E}$ instead of $\boldsymbol{T}$ to quantify the entanglement entropy.

Secondly, assume the compound relation of word vectors in $V_Q$(or $V_A$) are considered in the $k$-order ($k < b$), the dimension of $\mathcal{H}^Q$ is $a \times m$ ($am$) and $\mathcal{H}^A$ is $C_b^k \times m^k$ ($C_b^k m^k$). E.q. 4 can be rewritten as follow:

$$\boldsymbol{\psi}_S = \sum_{i=1}^{am} \sum_{j=1}^{C_b^k m^k} \boldsymbol{T}_{i,j} \boldsymbol{\phi}_i^Q \otimes \boldsymbol{\phi}_j^A \tag{13}$$

---

[6]$C_n^m$ represents the number of combinations of $m$ elements taken from $n$ different elements($m \leq n$)

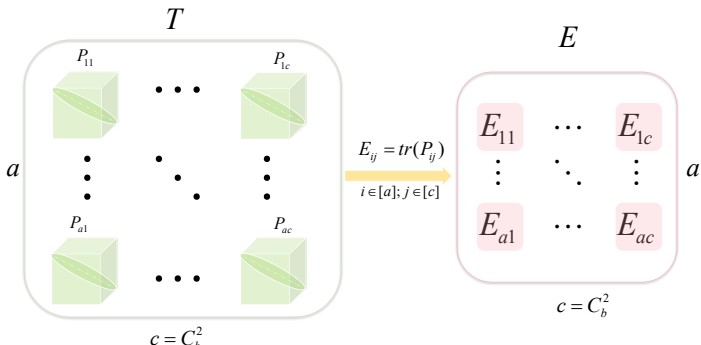

Figure 3: Assuming the compound relation of word vectors in $V_Q$ with $a$ words (or $V_A$ with $b$ words) are considered in the 2-order which represents $c$ ($c = C_b^2$) compound semantics of any two-word vectors in $V_A$. The relation between the attention matrix $E$ and the matrix $T$: $P_{ij}$ which can be regarded as a 3-order tensor is the block matrix from the matrix $T$, $E_{ij}$ is equal to the trace of matrix $P_{ij}$ ($i \in [a]; j \in [c]$).

where $\{\phi_i^Q\}_{i=1}^{am}$ and $\{\phi_j^A\}_{j=1}^{C_b^k m^k}$ are base vectors for $\mathcal{H}^A$ and $\mathcal{H}^Q$. The $T$ is a matrix of $am \times C_b^k m^k$. Divide the $T$ into $a \times C_b^k$ block matrices $\{P_{ij} \in \mathbb{R}^{m \times m^k}\}$ ($i \in [a]; j \in [C_b^k]$), which can be written as $P_{ij} = w_i^Q \otimes w_j^A$, $w_j^A$ ($w_j^A \in \mathbb{R}^{m^k}$) represents the compound semantics of any $k$ word vectors in $V_A$. Matrix $P_{ij}$ can be regarded as a $k$-order tensor $P_{t_1 t_2 \dots t_k}$ ($P_{t_1 t_2 \dots t_k} \in \mathbb{R}^{m \times m \times \dots \times m}$), each element $E_{ij}$ ($i \in [a]; j \in [C_b^k]$) of the attention matrix $E$ is equal to the trace of the $k$-order tensor, i.e., $E_{ij} = <w_{t_1}^Q, w_{t_2}^A \cdot w_{t_3}^A \dots w_{t_k}^A>$ ($t_1 \in [a], t_2 \neq t_3 \neq \dots \neq t_k$ and $t_2, t_3, \dots t_k \in [b]$), which can be written as:

$$E_{ij} = trace(P_{ij}) \tag{14}$$

Thus, we use the attention matrix $E$ instead of $T$ to quantify the entanglement entropy. Thirdly, assume the compound relation of word vectors in $V_Q$(or $V_A$) are considered in the $(k+1)$-order ($k < b$), the dimension of $\mathcal{H}^Q$ is $a \times m$ ($am$) and $\mathcal{H}^A$ is $C_b^{k+1} \times m^{k+1}$ ($C_b^{k+1} m^{k+1}$). E.q. 4 can be rewritten as follow:

$$\psi_S = \sum_{i=1}^{am} \sum_{j=1}^{C_b^{k+1} m^{k+1}} T_{i,j} \phi_i^Q \otimes \phi_j^A \tag{15}$$

where $\{\phi_i^Q\}_{i=1}^{am}$ and $\{\phi_j^A\}_{j=1}^{C_b^{k+1} m^{k+1}}$ are base vectors for $\mathcal{H}^A$ and $\mathcal{H}^Q$. The $T$ is a matrix of $am \times C_b^{k+1} m^{k+1}$. Divide the $T$ into $a \times C_b^{k+1}$ block matrices $\{P_{ij} \in \mathbb{R}^{m \times m^{k+1}}\}$ ($i \in [a]; j \in [C_b^{k+1}]$), which can be written as $P_{ij} = w_i^Q \otimes w_j^A$, $w_j^A$ ($w_j^A \in \mathbb{R}^{m^{k+1}}$) represents the compound semantics of any $(k+1)$ word vectors in $V_A$. Matrix $P_{ij}$ can be regarded as a $(k+1)$-order tensor $P_{t_1 t_2 \dots t_{(k+1)}}$ ($P_{t_1 t_2 \dots t_{(k+1)}} \in \mathbb{R}^{m \times m \times \dots \times m}$), each element $E_{ij}$ ($i \in [a]; j \in [C_b^{k+1}]$) of the attention matrix $E$ is equal to the trace of the $(k+1)$-order tensor, i.e., $E_{ij} = <w_{t_1}^Q, w_{t_2}^A \cdot w_{t_3}^A \dots w_{t(k+1)}^A>$ ($t_1 \in [a], t_2 \neq t_3 \neq \dots \neq t_{(k+1)}$ and $t_2, t_3, \dots t_{(k+1)} \in [b]$), which can be written as:

$$E_{ij} = trace(P_{ij}) \tag{16}$$

Thus, we use the attention matrix $E$ instead of $T$ to quantify the entanglement entropy.

In summary, the equivalence relationship between the attention matrix $E$ and the entanglement properties matrix $T$ is satisfied under the 1-order condition, the 2-order condition, the $k$-order condition, and the (k+1)-order condition, which proves that the relationship is established. That is, attention and entanglement properties are equivalent under specific premise.

$\square$

