# OpenReview forum: "Leveraging Entanglement Entropy for Deep Understanding of  Attention Matrix in Text Matching"
_ICLR.cc/2020/Conference — Reject_

### Official Review · AnonReviewer2 · 2019-10-23
**Official Blind Review #2**

**Rating:** 1

**Review:**

The paper deals with the understanding of deep learning under a physical point of view, related to entanglement entropy.
As far as i understand the paper explains how the computation of the entanglement entropy may be performed and how this measure may be used to design the neural network architecture.

I was pretty much interested by the paper but I was unfortunately not able to follow the « theoretical » part of the paper, which in my opinion is not well introduced. The paper cites papers from physical letters and physical reviews without introducing all necessary background. From this point of view the paper is not self content enough for the audience of the conference.
I don’t know what is the quantum many-body problem, never heard of StartEnd separation rank… mentioned in the first paragraph of the paper.

In addition to the fundamental contribution that i could not summarize well, the paper includes an experimental validation of the contribution on the question answering problem on benchmark datasets, showing very relevant results outperforming the baselines, which look like state of the art methods in the field.

Not sure to fully understand the experimental setting though. In particular i understand that the dataset are divided in isolated parts on which different models are learned. And averaged results are reported. But is it a fair comparison with the baselines ?

At the end the paper looks quite interesting and promising but in its current shape it doesn’t seem to me fully accessible to ICLR audience.

**Experience Assessment:**

I do not know much about this area.

**Review Assessment: Checking Correctness Of Derivations And Theory:**

I did not assess the derivations or theory.

**Review Assessment: Checking Correctness Of Experiments:**

I assessed the sensibility of the experiments.

**Review Assessment: Thoroughness In Paper Reading:**

I read the paper at least twice and used my best judgement in assessing the paper.

---

> ### Author Response · Authors · 2019-11-10
> **Authors' response to reviewer 2**
>
> We thank the reviewer for the time and feedback.
> 1 We would like to in the experiment, we did not compute an average evaluation result on two sub-datasets. Instead, we combine the QA-pair’s matching score files from the two sub-datasets into one file and calculate the MAP and MRR of the entire dataset based on this file. This is not different from the general practice of evaluating the entire dataset. Detailed experiments are described in Section 4.2.
>
> 2 About the quantum many-body problem, in Physics, Quantum Many-body Wave Function (QMWF) can model the interaction among many particles and the associated basis vectors. In the language scenario, by considering a word as a particle, different meanings (or latent/embedded concepts) as different basis vectors, the interaction among words (or word meanings) can be modeled by the tensor product of basis vectors, via the many-body wave function.
> .

---

### Official Review · AnonReviewer1 · 2019-10-23
**Official Blind Review #1**

**Rating:** 1

**Review:**

In this paper, a method for leveraging entanglement entropy for understanding attention matrices is proposed. Specifically, the paper aims at solving two problems: 1) to study the theoretical analysis of entanglement entropy for the matching of two objects (question-answering pairs), and (2) to qualitatively calculate the matching matrix. The introduced approach is based on fundamental connections between the entanglement entropy and the attention matrix. The main goal of the paper is to show that a low-dimensional attention matrix can be derived from a high-dimensional matching matrix. Results are shown for a text matching task on two datasets (TREC-QA, YAHOO-QA).

My main concerns with this paper are that the approach is not well described and that the proposed contribution appears quite narrow. Overall it is not clear what the actual contribution of this work really is. Several sections of the paper appear convoluted and could be more concise; the text is often written unnecessarily complicated which makes it hard to follow and to comprehend details. While the experiments seem sound the overall improvement of 2.9% compared to SOTA work appear rather shallow. Therefore, in its current state I cannot recommend accepting this paper.

Detailed comments:

- The abstract of the paper is very cryptic and the motivation for the proposed approach is not sufficiently described.
- The paper is not well-written and difficult to follow. Several sentences and sections are left too unclear. E.g. in the introduction sentences like "... but such an indicator only reflects the intricate correlation structures of each single input object (e.g., an image or a text)" or "This is due to the fact that the tensor product occurs in the quantum many-body function for representing the image and text " are confusing as they come without much context.
- It is not clear what is meant by "matching problem".
- A few sentences in the introduction are exact copies of the abstract, which makes the text appear redundant. It would help to clearly state what the contribution of this work is, instead of repeating sentences.
- It is not clear what is meant by "relatively-deeper" and "relatively-shallower" layers.
- The text shows several spelling and grammar issues (such as "for the more complex the inputs").
- Several sentences don't make sense and are difficult to read (e.g. "Since our work is mainly for the text matching task of a sentence pair, we briefly introduce a recent Quantum Many-body Wave Function inspired Language Modeling").
- The notation and equations in Section 2 are mostly common knowledge and could be moved to the appendix.
- In Section 3.1. what is meant by "subsystem in deep neural networks"? Later in the text in becomes more clear. So it would be helpful to rearrange the text.
- "probability amplitude distributions " is not clear.
- Sentences like "which often correspond to the important information hidden in the matrix" should be accompanied with a reference.
- Section 3.2 is titled "Network Design Based on Entanglement Entropy", but the section does not actually describe a network architecture, but instead just describes how to obtain the attention matrix and the sample differences.
- Section 4.1 (first paragraph): it would help to discuss the connection of many-body wave functions to represent questions and answer sentences as two subsystems more clearly and earlier in the text.
- Section 4.2 (second paragraph) appears quite repetitive. Many sentences have been used in previous sections of the text.
- The results and discussions shown in Sections 4.4 and 4.5 are interesting and seem sound. However, it is not clear what exactly are the "adaptive settings for kernels" (e.g. in Figure 2)
- Limitations of the approach are not sufficiently discussed.
- It is not clear what is meant by "we will investigate the entanglement entropy under high-order conditions" in the conclusion.

**Experience Assessment:**

I have read many papers in this area.

**Review Assessment: Checking Correctness Of Derivations And Theory:**

I assessed the sensibility of the derivations and theory.

**Review Assessment: Checking Correctness Of Experiments:**

I assessed the sensibility of the experiments.

**Review Assessment: Thoroughness In Paper Reading:**

I read the paper at least twice and used my best judgement in assessing the paper.

---

> ### Author Response · Authors · 2019-11-15
> **Authors' response to reviewer 1**
>
> We thank the reviewer for the detailed suggestions. We will revise our manuscript according to your suggestions on the paper’s presentation.

---

### Official Review · AnonReviewer4 · 2019-10-29
**Official Blind Review #4**

**Rating:** 3

**Review:**

This work extends the Quantum Many-body Wave Function inspired language model (QMWF-LM) of Zhang et al by proposing some quantum entanglement entropy computation to separate the data into long range correlation and short range correlation. The authors report improved results on the TREC-QA and YAHOO-QA datasets.

I am not an expert on Quantum physics, hence I am unable to judge the merits of the quantum entanglement approach proposed in the paper. However, the work that this paper builds on - the "QMWF-LM of Zhang et al" - doesn't seem to have been vetted by proper peer review in either an ML conference or journal. I am also skeptical of the quantum terminology introduced in the paper and the experiments are reported on only two QA datasets - TREC and YAHOO which aren't super standard. If Quantum inspired language models are the next big progress in language modeling, I would like to see more experiments on some language modeling datasets such as LM1B (Chelba et al), Wikitext-2/103 (Merity et al) etc. Besides, this approach should also work for SQuAD, GLUE, SuperGLUE and all the other established NLP benchmarks that benefit from improved language modeling capabilities.

I am also skeptical of the progress in TREC-QA, the state-of-the-art claimed by the authors is Kamath et al which uses an RNN + pre-attention. A stronger and more modern baseline would be something like BERT (Devlin et al.) or any of the subsequent improvements to it.

This paper would benefit from a clearer exposition minus the quantum mechanics jargon and from experiments on the above stated benchmarks to be more convincing.

**Experience Assessment:**

I have read many papers in this area.

**Review Assessment: Checking Correctness Of Derivations And Theory:**

I assessed the sensibility of the derivations and theory.

**Review Assessment: Checking Correctness Of Experiments:**

I assessed the sensibility of the experiments.

**Review Assessment: Thoroughness In Paper Reading:**

I made a quick assessment of this paper.

---

> ### Author Response · Authors · 2019-11-10
> **Authors' response to reviewer 4**
>
> We thank the reviewer for the time and feedback; Our responses are as follows.
> 1 QMWF-LM is peer-reviewed in CIKM (Zhang et al. CIKM 2018), where QMWF-LM is an abbreviation for A Quantum Many-body Wave Function Inspired Language Modeling.
> 2 There is no problem with the quantum terminology in our paper. We have already explained this issue in the first paragraph of Section 2. The formulation in this paper makes it easier for the reader to understand.
> 3 In the experiment, the TREC-QA and YAHOO-QA are two typical Q&A datasets on question answering tasks.
> 4 Quantum Many-body Wave Function Inspired Language Modeling reveals the inherent necessity of using Convolutional Neural Network (CNN) in quantum-inspired language modeling. In our experiments, we compare our methods with two state-of-the-art CNN-based QA methods.
> 5 One of the main contributions of this paper is to prove the equivalence between the Attention Matrix and quantum entanglement under certain conditions.

---

### Decision · Program_Chairs · 2019-12-19

**Decision:**

Reject

**Comment:**

This paper advocates for the application of entanglement entropy from quantum physics to understand and improve the inductive bias of neural network architectures for question answering tasks. All reviewers found the current presentation of the method difficult to understand, and as a result it is difficult to determine what exactly the contribution of this work is. One suggestion for improving the manuscript is to minimize the references to quantum entanglement (where currently is it asserted without justification that entanglement entropy is a relevant concept for modeling question-answering tasks). Instead, presenting the method as applications of tensor decompositions for parameterizing neural network architectures would make the work more accessible to a machine learning audience, and help clarify the contribution with respect to related works [1].

1. http://papers.nips.cc/paper/8495-a-tensorized-transformer-for-language-modeling.pdf